# Omnidirectional Triboelectric Nanogenerator for Wide-Speed-Range Wind Energy Harvesting

**DOI:** 10.3390/nano12224046

**Published:** 2022-11-17

**Authors:** Qiman Wang, Wenhao Li, Kun Wang, Yitao Liao, Junjie Zheng, Xiongtu Zhou, Jianpu Lin, Yongai Zhang, Chaoxing Wu

**Affiliations:** 1College of Advanced Manufacturing, Fuzhou University, Quanzhou 362251, China; 2College of Physics and Information Engineering, Fuzhou University, Fuzhou 350108, China; 3Fujian Science and Technology Innovation Laboratory for Optoelectronic Information of China, Fuzhou 350108, China

**Keywords:** triboelectric nanogenerators, core–shell structure, wind energy harvesting, electronics

## Abstract

The environmentally friendly harvesting of wind energy is an effective technique for achieving carbon neutrality and a green economy. In this work, a core–shell triboelectric nanogenerator (CS-TENG) for harvesting wind energy is demonstrated and the device structure parameters are optimized. The core–shell structure enables the CS-TENG to respond sensitively to wind from any direction and generate electrical output on the basis of the vertical contact–separation mode. A single device can generate a maximum power density of 0.14 W/m^3^ and can power 124 light-emitting diodes. In addition, wind energy can be harvested even at a wind speed as low as 2.3 m/s by paralleling CS-TENGs of different sizes. Finally, a self-powered water quality testing system that uses the CS-TENG as its power supply is built. The CS-TENG exhibits the advantages of a simple structure, environmentally friendly materials, low cost, and simple fabrication process. These features are of considerable significance for the development of green energy harvesting devices.

## 1. Introduction

Energy shortage has always been a popular topic of concern worldwide [1,2,3]. To alleviate the energy crisis and reduce overreliance on chemical batteries, humans are constantly searching for alternatives to nonrenewable energy sources. The triboelectric nanogenerator (TENG) technology proposed by Wang in 2012 has been proven to be an emerging distributed energy harvesting technology with broad application prospects [4,5,6,7,8,9]. The primary function of TENGs is to efficiently convert various forms of small and irregular mechanical energy in the surrounding environment into electrical energy to power distributed sensors and small electronic devices. Among such forms of mechanical energy, wind energy has received extensive attention because of its unique advantages, including wide distribution and large storage capacity, making it an ideal natural environment energy source.

In recent years, TENGs for harvesting wind energy have achieved significant progress in terms of structures and materials [10,11,12,13,14,15,16,17,18]. Typical TENG structures for efficient wind energy harvesting can be generally classified into the wind cup/windmill type [19,20,21,22] and the flag type [23,24]. The wind cup/windmill type is mostly used in rotating-type TENGs. That is, wind cups or blades rotate under the driving of the wind, converting the linear motion of the air into the rotary relative motion between friction layers. Wind cup/windmill type TENGs typically operate in freestanding triboelectric-layer mode [25,26,27,28] and lateral sliding mode [29,30]. Wang et al. demonstrated a dual-rotor TENG fabricated using wind cups; this TENG can stably harvest wind energy within a wind speed range of 2.6–16 m/s [31]. Zhang et al. proposed a windmill hybrid nanogenerator that can harvest wind energy down to 1.8 m/s with a peak power of 0.97 mW [32]. For flag-type TENGs, a pressure difference is formed on the two surfaces of the flag when wind blows on the flag, causing the flag to swing. Thus, the linear motion of the air is translated into the periodic back-and-forth motion of the flag. Flag-type TENGs typically operate in the vertical contact–separation and single-electrode modes. Xu et al. constructed a flag-type TENG with a maximum current of 6.8 μA and a peak output power of 36.72 mW at a wind speed of 7.5 m/s [33]. Meanwhile, friction materials for harvesting wind energy have also gradually developed from polymers into environmentally friendly materials. For example, Ma et al. used wheat straw as a friction material, and the output voltage of the device could reach 250 V [34]. Feng et al. fabricated TENG using fresh leaves; the device’s open-circuit voltage (V_OC_) and short-circuit current (I_SC_) were 430 V and 15 μA, respectively [35]. Although TENGs have achieved significant progress in harvesting wind energy, the room for improvement in practical applications is still considerable. TENGs with new structures should be developed to harvest wind energy efficiently, and eco-friendly friction materials with a high charge density should be developed. Finally, the application scenarios of TENGs for wind energy harvesting should be expanded.

Here, we present a lightweight, low-cost, and eco-friendly core–shell TENG (CS-TENG). This CS-TENG consists of a small cuboid box inside a larger cuboid box. The linear motion of air is converted into vertical contact–separation between the two boxes. Thus, wind energy can be converted into electricity through contact electrification and electrostatic induction processes. Structural parameters, such as the size and weight of the CS-TENG, are optimized to further improve output performance. Device performance with one and two pairs of friction surfaces is discussed. The CS-TENG is demonstrated to harvest wind energy from arbitrary directions, and a maximum V_OC_ of 180 V can be obtained in either the vertical or horizontal direction. In addition, CS-TENGs with different structural parameters are connected in parallel to collect wind energy within a broad range of speeds (2.3–8.8 m/s). Finally, we establish a self-powered water quality detection system using CS-TENGs as a power source to detect water quality pH value. This work proposes a TENG that cannot only accelerate the development of the green economy, but can also provide an effective method for harvesting wind energy. It is widely accepted that the TENG with hydrogels has high flexibility, adjustable mechanical properties, and excellent electrochemical properties for wind energy harvesting [36]. One of the starting points of this work is to make the fabrication of wind-harvesting TENGs become easy. The paper-based CS-TENG has advantages, including that (1) the materials are easy to obtain and low cost; (2) the fabrication process is simple; and (3) the device is environmentally friendly.

## 2. Experimental Section

### 2.1. Fabrication of the CS-TENG

Two boxes with dimensions of 10 × 10 × 30 cm^3^ and 7 × 7 × 24 cm^3^ are made using cardboard (1 mm thick). The smaller box hangs from inside the larger box, and they do not touch each other when they are at rest. When an external shock is applied, the outer surface of the small box can come in contact the inner surface of the large box. Nylon and polytetrafluoroethylene (PTFE) with a thickness of 0.02 mm are used as the positive and negative friction materials, respectively. Copper foil with a thickness of 0.06 mm is used as the electrode. Copper/nylon stacked film (10 × 30 cm^2^) is attached onto the inner surface of the larger box. Copper/PTFE (7 × 24 cm^2^) is attached onto the outer surface of the smaller box.

### 2.2. Measurement

A digital oscilloscope (RIGOL, DS2302A, (Suzhou, China) is used to measure the output voltage, while an electrometer (KEITHLEY, 6514B, (Beaverton, OR, USA) is used to measure the output current. An anemometer (GM8902, (Lexiang Electronics, Guangzhou, China) is used to measure wind speed.

## 3. Results and Discussion

### 3.1. Structural Design and Working Principle of the CS-TENG

The potential application scenario of CS-TENG is illustrated in Figure 1a, wherein the lantern-shaped CS-TENG is hung on a street lamp, allowing it to swing with the wind. The CS-TENG consists of two boxes with different sizes, as shown in Figure 1b. The inner box acts as the “core”, while the outer box acts as the “shell”. The smaller box hangs inside the larger box, and they do not touch each other when they are at rest. When an external shock is applied, the outer surface of the small box can come into contact with the inner surface of the big box. The back-and-forth collision of the inner and outer boxes leads to the periodic contact and separation of the two friction surfaces, realizing the conversion of wind energy into electrical energy. Copper foil is used as the electrodes. The PTFE film used as the negative friction layer is attached onto the outer surface of the smaller box, while the nylon film used as the positive friction material is attached onto the inner surface of the larger box. The materials used to fabricate the CS-TENG are lightweight and inexpensive. As shown in Figure 1c, the weight of the 7 × 7 × 24 cm^3^ rectangular box made of the 1 mm thick cardboard is about 40 g. The overall manufacturing cost of the device is low and the device is suitable for mass production.

The working mechanism of the CS-TENG is the periodic contact and separation between the inner and outer boxes, and its theoretical basis is the coupling between electrification and electrostatic induction. The detailed charge transfer and circuit connection processes are schematically illustrated in Figure 1d. The inner and outer boxes have four pairs of friction surfaces, thus the power generation process is the same when the boxes come into contact and separate. We use two pairs of opposite friction surfaces (defined as pair-I and pair-II) to illustrate the working process of the CS-TENG. When the CS-TENG vibrates because of blowing wind, contact and friction occur between the PTFE film (inner box) and the nylon film (outer box). In accordance with the triboelectric coefficients of the materials, the PTFE surface is negatively charged and the nylon membrane is positively charged. In the beginning, we assume that the inner box is in the left (Figure 1d,I). Therefore, the copper/PTFE and copper/nylon of pair-I are in contact, whereas the copper/PTFE and copper/nylon of pair-II are separated. The distribution of charges is illustrated in Figure 1d,I. Subsequently, the inner box starts to move to the right, creating a potential difference between the copper electrodes of pair-I and pair-II that drives the flow of electrons. For pair-I, electrons move from the inner box to the outer box; for pair-II, electrons move from the outer box to the inner box, as illustrated in Figure 1d,II. Current flowing through an external electric load has the same direction owing to the use of rectifier bridges. As shown in Figure 1d,III, pair-I and pair-II have the same charge distribution state when the inner box is moved to the middle position. When the inner box continues to move to the right, the electrons of pair-I move from the inner box to the outer box, while the electrons of pair-II move from the outer box to the inner box, as illustrated in Figure 1d,IV. Notably, the potentials of pair-I and pair-II are different during the movement of the inner box from left side to right side, thus two current pulses are obtained. In the same manner, when the inner box moves from the right side to the left side, the charge transfer processes for pair-I and pair-II are reversed. However, the direction of the current flowing through the external electric load does not change because of the presence of rectifier bridges. Two contact–separation processes occur in one complete motion cycle and four current pulses will be generated, as discussed in the following section.

### 3.2. Structural Optimization of the CS-TENG

For the CS-TENG, the dimensions of the inner and outer boxes exert a significant effect on the device output performance. On the one hand, the size of the box will affect the relative movement speed of the box under a certain wind speed. On the other hand, the size of the box determines the contact area of the friction layer. Therefore, finding suitable device structure parameters to improve the output performance is necessary. First, the influence of the length and width of the inner box on output performance is studied. An outer box with a fixed size (10 × 10 × 30 cm^3^) is used. Several inner boxes with the same height (28 cm) and different basal areas (3 × 3, 4 × 4, 5 × 5, 6 × 6, 7 × 7, and 8 × 8 cm^2^) are made. As shown in Figure 2a,b, V_OC_ and I_SC_ increase with an increase in the inner box basal area, because the contact area increases with the basal area of the inner box. When the substrate area is 7 × 7 cm^2^, the maximum V_OC_ (140 V) and the maximum I_SC_ (4.5 μA) can be obtained. However, electrical output decreases once the inner box basal area is further increased. The reason for this phenomenon is that the distance between the inner and outer boxes decreases when they are in a separated state when the basal area of the inner box is too large. Therefore, the potential difference between the two electrodes is not sufficiently large, reducing the amount of charge transferred. The bottom area of the inner box increases from 8 × 8 cm^2^ to 9 × 9 cm^2^. It can be found that the V_OC_ and I_SC_ decrease with the increase in the inner box bottom area (Appendix A).

In addition, we investigate the effect of inner box height on device output performance. We make an outer box with a fixed size (10 × 10 × 30 cm^3^) and several inner boxes with the same bottom (7 × 7 cm^2^) and different heights (18, 20, 22, 24, 26, and 28 cm). As shown in Figure 2c,d, V_OC_ and I_SC_ increase with an increase in the inner box height. The reason for this phenomenon is that the contact area increases with the height of the inner box. When inner box height is 24 cm, V_OC_ and I_SC_ reach a maximum value of 160 V and 4.7 μA, respectively. However, electrical output decreases once the inner box height is further increased. The inner box is suspended inside the outer box; hence, an increase in the height of the inner box will inevitably lead to a decrease in the distance between the top of the inner box and the top of the outer box. Therefore, the contact area of the inner and outer boxes decreases instead when the height of the inner box is too large.

Finally, we investigate the effect of the inner box weight on the output performance of the device, because the inner box weight will affect the relative motion of the inner and outer boxes. As shown in Figure 2e,f, V_OC_ and I_SC_ gradually increase with an increase in the inner box weight. The device achieves the best output (V_OC_ = 180 V, I_SC_ = 4.9 μA) when the inner box weight is 60 g. The reasons that the weight of the inner box affect the electrical output of the device are as follows. (1) When the inner box is lighter, the swing amplitudes of the inner and outer boxes are relatively close. Consequently, the probability of contact is small and the separation distance is also small, resulting in lower output voltage and current. (2) When the weight of the inner box increases, the swing amplitude and swing frequency of the inner and outer boxes are inconsistent. Therefore, the probability of contact between the two boxes increases and the speed of contact–separation is faster. Thus, the output of the device is enhanced. (3) When the weight of the inner box is too large, the inner box tends to be in a static state. In such a case, the speed of the contact–separation of the two boxes is reduced, decreasing output performance. In accordance with the preceding discussion, the optimized CS-TENG has an outer box of 10 × 10 × 30 cm^3^ and an inner box of 7 × 7 × 24 cm^3^. It can be found that the output of the device is unstable. The reason is that the active driving force is always changing when driven by wind. However, we can use energy storage devices to store the electrical energy and then power electronic devices. Thus, the unstable output electricity would not affect the practical application.

### 3.3. Performance of the CS-TENG

While discussing the working process of the device, we mention that, if two friction pairs are used, then two contact separation processes will occur in one working cycle (Figure 1d). Here, we demonstrate the experimental results when one friction pair and two friction pairs are used. First, only one friction pair (A1 and A2) is used, as schematically illustrated in Figure 3a. When the CS-TENG is subjected to an external force, periodic contact–separation occurs on the surfaces of A1 and A2, and peak V_OC_ and I_SC_ reach 180 V and 5 μA, respectively, as shown in Figure 3b,c. Only two voltage (or current) pulses are found in each contact–separation cycle. The first pulse is generated from the contact between the A1 and A2 surfaces. The second pulse is generated from the separation of the A1 and A2 surfaces. Subsequently, another friction pair (B1 and B2) is added to the CS-TENG. In this case, the device has two friction pairs, as illustrated in Figure 3d. The output V_OC_ and I_SC_ of the device with two friction pairs are illustrated in Figure 3e,f. The peak values of V_OC_ and I_SC_ are similar to those of the device with only one friction pair. Moreover, the CS-TENG with two friction pairs generates more electrical signal pulses per unit time. Four voltage (or current) pulses occur in each contact–separation cycle. As shown in Figure 3e, the first pulse is generated from the contact between the A1 and A2 surfaces and the third pulse is generated from the separation of the A1 and A2 surfaces. The second pulse is generated from the separation of the B1 and B2 surfaces and the fourth pulse is generated from the contact between the B1 and B2 surfaces.

To further compare the electrical output performance of the device with different numbers of friction pairs, we investigate the peak power of the two devices. The peak power of the CS-TENG with one friction pair is 0.42 mW and the best resistance is about 4.2 × 10^8^ Ω (Figure 3g). For the device with two friction pairs, a peak power of 0.13 mW can be obtained at about 4 × 10^7^ Ω, as shown in Figure 3h. The reason for the drop in peak power is the change in the internal resistance of the device. Power reaches its maximum value when the external and internal resistances are equal in a circuit [37,38]. When two friction pairs are connected in parallel through the rectifier bridge, the overall internal resistance of the device is reduced, along with the impedance that corresponds to the maximum peak power. Therefore, the peak power of the CS-TENG with two friction pairs is reduced compared with that of the device with only one friction pair. However, a reduction in peak power does not equate to a reduction in total output power. The output of the CS-TENG is a pulse, thus output power is not only affected by the peak power of the pulse, but also by the number of pulses per unit time. As mentioned earlier, the use of multi-friction pairs can increase the output pulse number per unit time. Therefore, the overall output of the CS-TENG with two friction pairs should be higher than that of the device with only one friction pair. To prove this conclusion, we use the two devices to charge a 22 µF capacitor. As shown in Figure 3i, the capacitor’s voltage increases to 1 V after a charging time of 25 s when charged with the CS-TENG with a single friction pair. However, when the CS-TENG with two friction pairs is used, the capacitor’s voltage reaches about 2.1 V. The charging rate of the CS-TENG with two friction pairs is nearly two times that of the CS-TENG with a single friction pair, and the available output energy at the same time is about four times higher than the CS-TENG with a single friction pair. On the basis of the preceding analysis, the CS-TENG with two friction pairs exhibits better output performance than the CS-TENG with only one friction pair.

Our CS-TENG has four sides where friction layers can be placed. However, when only two pairs that faced each other are used, only wind energy whose wind direction is along the normal direction of the friction surface can be collected. When the friction layers are placed on the four sides of the CS-TENG, the CS-TENG can successfully harvest wind energy from all directions. Figure 4 illustrates the ability of the CS-TENG with four friction pairs to harvest wind energy at different wind blowing angles. Notably, the angles 0° and 90° are defined as wind direction along the normal direction of the first and second friction pairs, respectively, as shown in the inset of Figure 4a. When wind direction is along the 0° and 90° directions, V_OC_ is about 175 V. When wind direction is along the 30°, 45°, and 60° directions, V_OC_ is only about 62 V, as shown in Figure 4a,b. The variation trend of I_SC_ is consistent with that of V_OC_, as depicted in Figure 4c,d. The measured I_SC_ at 0° and 90° is about 8 μA, which is higher than the I_SC_ at 30°, 45°, and 60°. The electrical output of the CS-TENG in the horizontal (0° and 180°) or vertical (90° and 270°) directions is higher than those in the other directions, because the actual contact area of the inner and outer boxes is larger in these two directions than in the other directions.

The peak power generated by the CS-TENG at different vibration angles is discussed. When the CS-TENG works in the vibration directions of 0° and 90°, peak power can reach 0.42 mW, which is about two times that of the 30° and 60° directions, as shown in Figure 4e. We use this CS-TENG to charge a 22 μF capacitor and compare the charging rates at different vibration angles. The charging rates are similar when operating in the 0° and 90° directions, and V_OC_ reaches 1.52 V and 1.46 V, respectively, at a charging time of 20 s. However, when the CS-TENG is operated at 30°, V_OC_ is only 0.5 V during the same operating time, as shown in Figure 4f. In summary, charging rates vary at different vibration angles owing to the difference in contact area. In the actual application process, the real-time electrical output of the device will also change randomly because of the randomness of the vibration direction.

To allow the CS-TENG to operate within a wide range of wind speeds, CS-TENGs of different sizes are connected in parallel. The circuit is designed for paralleling CS-TENGs of different sizes, as shown in Figure 4g. Each unit is individually rectified and then connected in parallel to power electronics. The relationship between the weight of CS-TENGs of different sizes and their turn-on speeds (defined as the wind speed that can produce a measured electrical output) is investigated, as shown in Figure 4h. The results indicate that the turn-on speed of CS-TENGs increases with the increasing weight. Therefore, when the external wind speed is unstable, CS-TENGs of different sizes will automatically adjust themselves after being connected in parallel to obtain a wider wind speed range. We compare the charging rate between a single CS-TENG (outer box: 10 × 10 × 30 cm^3^, inner box: 7 × 7 × 24 cm^3^) and a CS-TENG group. For the CS-TENG group, two CS-TENGs are connected in parallel (outer box 1: 10 × 10 × 30 cm^3^, inner box 1: 7 × 7 × 24 cm^3^; outer box 2: 8 × 8 × 20 cm^3^, inner box 2: 6 × 6 × 16 cm^3^). With the same charging time (35 s), the single CS-TENG and the CS-TENG group can achieve a V_OC_ of 3 V and 4.1 V, respectively, as shown in Figure 4i.

To demonstrate the practical use of the CS-TENG in daily life, we study the effect of different wind speeds on the output performance of the CS-TENG. The relationship between wind speed and the V_OC_ of the CS-TENG is illustrated in Figure 5a. Output voltage increases as wind speed increases. Actual wind speed and direction are random, thus the electrical output of the CS-TENG is also random, requiring storage devices, such as capacitors, to store electrical energy. The charging voltage of the capacitor (22 μF) at different wind speeds is shown in Figure 5b. The charging rate of a 22 μF capacitor increases with an increase in wind speed. This phenomenon is ascribed to the oscillation amplitude of the CS-TENG becoming larger as wind speed increases, helping improve contact between the inner and outer boxes. At a wind speed of 9 m/s, charging a 22 μF capacitor from 0 V to 3 V takes 35 s. This capacitor can drive a micro calculator to work for 6 s, as shown in Figure 5c. Future applications of the CS-TENG are presented in Figure 5d, where multiple applications, including self-powered electronics and water quality detection, can be realized by harvesting wind energy. Considering that the rain and snow environment will reduce the mechanical performance of paper cards, we can set up small rain shelters where CS-TENGs are suspended to reduce the adverse impact on the output performance. In addition, the electrical energy converted by the CS-TENG can power 124 LEDs, as shown in Figure 5e. We demonstrate a self-powered water quality detection system composed of the CS-TENG and a water quality pH test pen, which can indicate whether water quality meets the standard without an external power supply, as shown in Figure 5f. In addition, the performance comparison between our CS-TENG and the recently reported wind-harvesting TENG is presented in Table 1. V_OC_ and I_sc_ of our CS-TENG are at the high level and the peak power is large. In a word, our CS-TENG has good output performance for harvesting wind energy.

## 4. Conclusions

In summary, a CS-TENG for harvesting wind energy is demonstrated and the device structure parameters are optimized. The core–shell structure enables the CS-TENG to respond to wind from any direction. This optimized CS-TENG has a V_OC_ of 180 V with a maximum power density of 0.14 W/m^3^. By connecting CS-TENGs of different sizes in parallel, wind energy within a wide range of wind speeds can be harvested and the charging rate can be increased. Finally, a self-powered water quality testing system that uses the CS-TENG as a power supply is built to achieve water quality testing. The CS-TENG exhibits the advantages of a simple structure, low cost, and simple fabrication process, which are highly important for the development of green energy harvesting devices.

## Figures and Tables

**Figure 1 nanomaterials-12-04046-f001:**
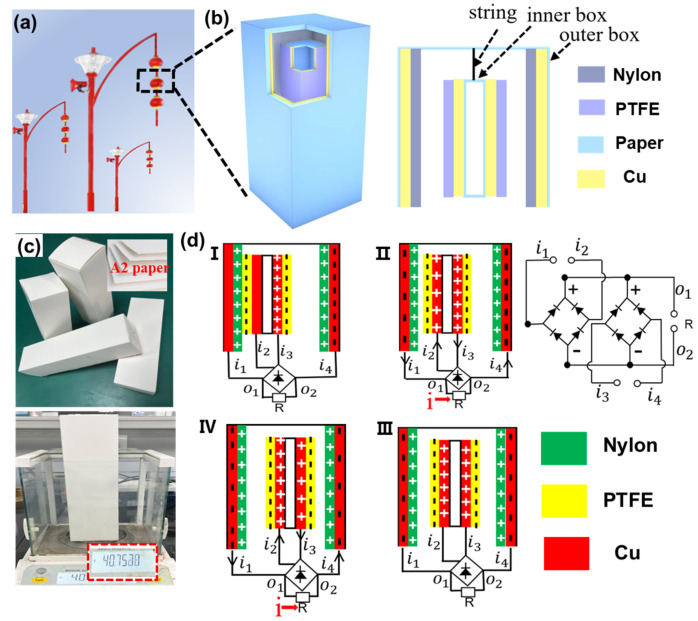
(**a**) Diagram of CS-TENG application scenarios. (**b**) Detailed structure of CS-TENG. (**c**) Photo images of the as–fabricated CS-TENG. (**d**) Schematic working process of CS-TENG. (**I**–**IV**) Charge distribution and current direction during the movement of the inner box from left to right. (**d-I**) The inner box is on the left. (**d-II**) The inner box is away from the left friction layer. (**d-III**) The inner box is in the middle position. (**d-IV**) The inner box is close to the right friction layer.

**Figure 2 nanomaterials-12-04046-f002:**
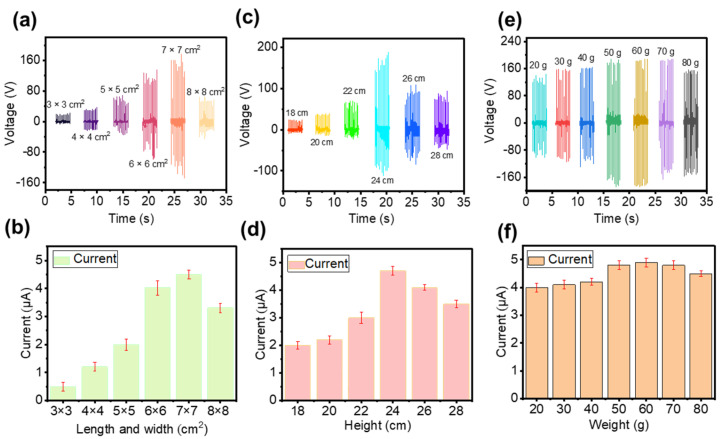
(**a**,**b**) V_OC_ and I_SC_ of the CS-TENG with different inner box lengths and widths. (**c**,**d**) V_OC_ and I_SC_ of the CS-TENG with different inner box heights. (**e**,**f**) V_OC_ and I_SC_ of the CS-TENG with different inner box weights.

**Figure 3 nanomaterials-12-04046-f003:**
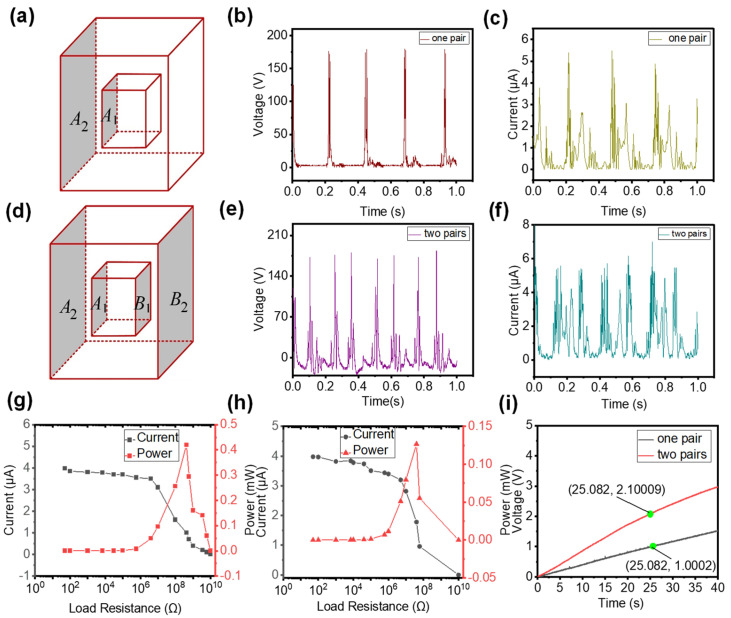
(**a**) Diagram of the CS-TENG with a single friction pair. (**b**) V_OC_ of the CS-TENG with a single friction pair. (**c**) I_SC_ of the CS-TENG with a single friction pair. (**d**) Diagram of the CS-TENG with two friction pairs. (**e**) V_OC_ of the CS-TENG with two friction pairs. (**f**) I_SC_ of the CS-TENG with two friction pairs. (**g**) Output peak power of the CS-TENG with a single friction pair. (**h**) Output peak power of the CS-TENG with two friction pairs. (**i**) Charging a 22 μF capacitor using the two types of CS-TENG.

**Figure 4 nanomaterials-12-04046-f004:**
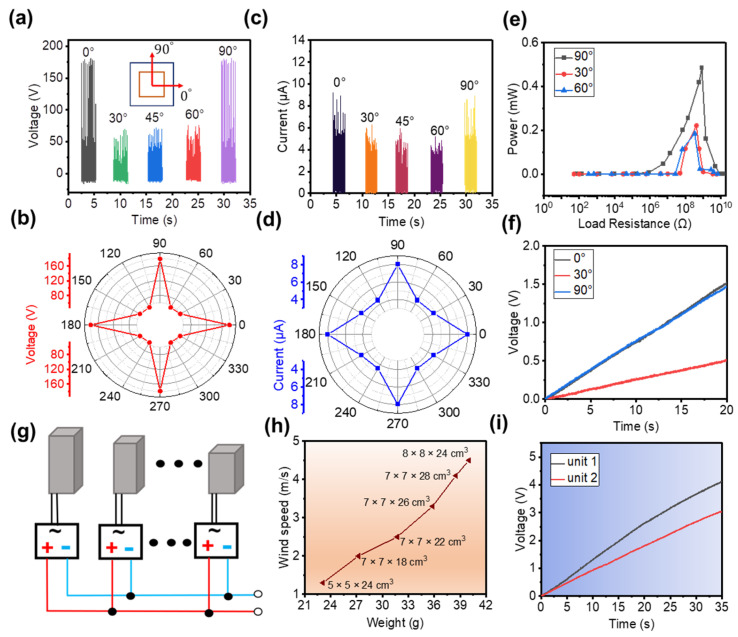
(**a**,**b**) V_OC_ of the CS-TENG at different wind angles. (**c**,**d**) I_SC_ of the CS-TENG at different wind angles. (**e**) Output power of the CS-TENG at different wind angles. (**f**) Charging a 22 μF capacitor at different wind angles. (**g**) Diagram of the parallel circuit with different CS-TENGs. (**h**) Turn-on wind speeds for CS-TENGs of different weight. (**i**) Charging performance of CS-TENGs with different numbers of integrated units.

**Figure 5 nanomaterials-12-04046-f005:**
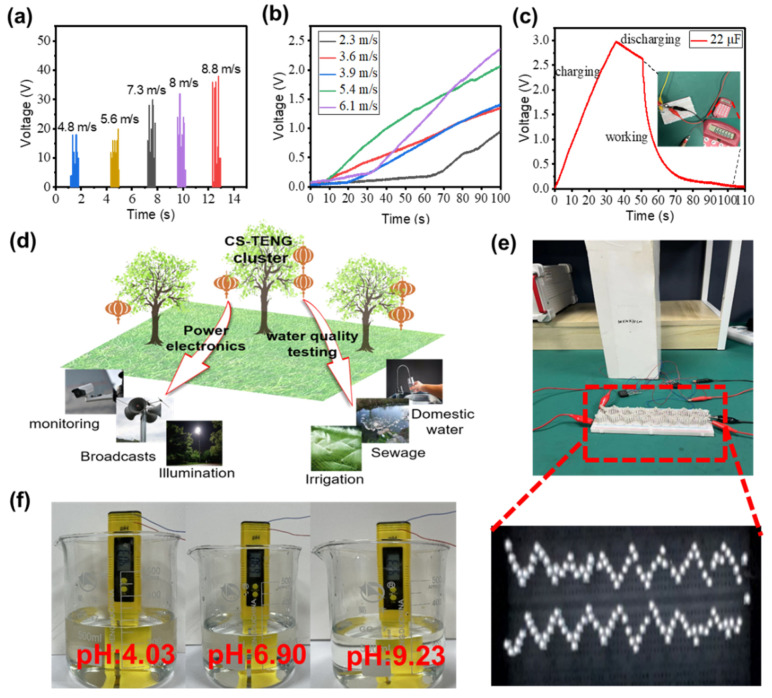
(**a**) V_OC_ of the CS-TENG at different wind speeds. (**b**) Charging performance at different wind speeds. (**c**) Charging and discharging processes of a capacitor to power a miniature calculator. (**d**) Prospect applications of the CS-TENG. (**e**) Photograph of the CS-TENG powering light-emitting diodes (LEDs). (**f**) Photograph of the self-powered pH test system.

**Table 1 nanomaterials-12-04046-t001:** Performance of CS-TENG and the reported wind-harvesting TENGs.

Type	V_OC_ (V)	I_SC_ (μA)	Power (mW)	Reference
Core–shell TENG	180	5.5	0.42	This work
Methyl-graphdiyne TENG	100	3.5	\	[12]
Pendulum TENG	56	0.5	\	[13]
Turbine-disk-type TENG	230	9	0.37	[14]
Angle-shaped TENG	64	2.5	0.2	[15]
Fluttering double-flag type TENG	70	6	\	[23]

## Data Availability

Not applicable.

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
