# Peer review of "Omnidirectional Triboelectric Nanogenerator for Wide-Speed-Range Wind Energy Harvesting"

_nanomaterials, 2022, doi:10.3390/nano12224046_

Round 1
Reviewer 1 Report
Wang et al. presented a TENG for wind energy harvesting applications. The work is interesting, but I am wondering if this work matching the scope of nanomaterial journal.
Few comments are given below..
1. It is not a paper based TENG, the working layer are PTFE and Nylon. So, the manuscript should be modified. Remove the words …. Paper based TENG.
2. Paper (cardboard) box are used as the supporting layer for the TENG. What about durability of the TENG? How to protect it from rain and thunderstorm while using outside as shown in Figure 1a and 5d?
3. A table to compare the performance with other reported wind harvester TENG should be provided.
Reviewer 2 Report
In this manuscript, the authors proposed a paper-based core-shell triboelectric nanogenerator (CS-TENG) to harvest wind energy. Wind from all directions could result in the electrical output of the CS-TENG by utilizing the contact-separation mode. The influence of structural parameters, such as size and weight, on the output performances, was further analyzed. Besides, the high output power density of this TENG could light 124 LED, which may demonstrate high potential in real life. I suggest the publication of this paper in Nanomaterials, if the author can address the following concerns.
1. In Figure 2, the author claimed that “The reason for this phenomenon is that the distance between the inner and outer boxes decreases when they are in a separated state when the basal area of the inner box is too large. Therefore, the potential difference between the two electrodes is not sufficiently large, reducing the amount of charge transferred.”, which was not so convincing. To further prove such a conclusion, the author should provide more control experiments. Besides, the output voltage in Figures a, c, and e was unstable, which may obstacle to the practical application. Furthermore, the current profile in Figure 2 b, d, and f should also contain error bars.
2. In Figure 1a, the author showed the potential applications of this core-shell TENG, which was hung on a street lamp to collect power from the wind. However, based on this manuscript, the TENG device demonstrated here was prepared by using papers. How the device can maintain its structure, shape, and configurations on a rainy day? If the author encapsulates the device with other hydrophobic materials, this would affect the power generation capacity of the device, which should be reassessed. The author should bring up a solution to demonstrate the feasibility of this project.
3. Most recently reported TENGs are based on flexible and stretchable materials, among which hydrogels with high mechanical compliance and ductility are proven as a good candidate (10.1002/adfm.202107437). Actually, the materials and structural configurations in this device were not new at all. The author should emphasize the novelty of this project, and compare this paper-based TENG with the one based on intrinsically stretchable hydrogels.
4. The author should summarize the output performances regarding this device and the published one to further the advantages of this manuscript.
5. The author claimed that the dimensions of the device affected the energy generation performances, thus, the output voltage and current may vary depending on the thickness of the cardboard and electrode categories. The author should also provide more data regarding these.

Round 2
Reviewer 1 Report
Accept in present form
Author Response
We thank the reviewer for the affirmation to our work.
Reviewer 2 Report
In this manuscript, the authors proposed a core-shell triboelectric nanogenerator (CS-TENG) with the merits of cost-effectiveness, simple device configurations, and environmental friendliness. Besides, this CS-TENG showed high voltage output and desirable power density. The concerns mentioned in the previous review were roughly addressed, however, the author should also find solutions for the following question before acceptance.
1. In the updated sentence “The paper-based CS-TENG has advantages including (1) Materials are easy to obtain and low cost; (2) Fabrication process is simple; (3) environmentally friendly.”, the three sentences formation after the colon should be the same. However, the first two were phrases, and the third was a phrase. The author should carefully re-check to avoid such small mistakes.

Author Response
We thank the reviewer for the affirmation to our work. We have corrected the sentence as: "The paper-based CS-TENG has advantages including: (1) Materials are easy to obtain and low cost; (2) Fabrication process is simple; (3) Device is environmentally friendly." (paragraph 2, page 2)